# Yukon River incision drove organic carbon burial in the Bering Sea during global climate changes at 2.6 and 1 Ma

Adrian M. Bender[1], Richard O. Lease[1], Lee B. Corbett[2], Paul R. Bierman[2], Marc W. Caffee[3], James V. Jones[1] and Doug Kreiner[1]

[1]U.S. Geological Survey, Alaska Science Center, 4210 University Drive, Anchorage, Alaska 99508, USA
[2]University of Vermont, Rubenstein School of the Environment and Natural Resources, 180 Colchester Avenue, Burlington, Vermont 05405, USA
[3]Purdue University, Department of Physics and Astronomy and Department of Earth, Atmospheric, and Planetary Sciences, 525 Northwestern Avenue, West Lafayette, Indiana 47907, USA

*Correspondence to*: Adrian M. Bender (abender@usgs.gov)

**Abstract.** River erosion affects the carbon cycle and thus climate by exporting terrigenous carbon to seafloor sediment and by nourishing $CO_2$-consuming marine life. The Yukon River-Bering Sea system preserves rare source-to-sink records of these processes across profound changes in global climate during the past five million years (Ma). Here, we expand the terrestrial erosion record by dating terraces along the Charley River, and explore linkages among previously published Yukon River tributary incision chronologies and Bering Sea sedimentation. Cosmogenic $^{26}Al/^{10}Be$ isochron burial ages of Charley River terraces match previously documented central Yukon River tributary incision from 2.6 to 1.6 Ma during Pliocene–Pleistocene glacial expansion, and at 1.1 Ma during the 1.2–0.7 Ma mid-Pleistocene climate transition. Bering Sea sediments preserve 2–4-fold rate increases of Yukon River-derived continental detritus, terrestrial and marine organic carbon, and silicate microfossil deposition at 2.6–2.1 Ma and 1.1–0.8 Ma. These tightly coupled records demonstrate elevated terrigenous nutrient and carbon export and concomitant Bering Sea productivity in response to climate-forced Yukon River incision. Carbon burial related to accelerated terrestrial erosion may contribute to $CO_2$ drawdown across the Pliocene–Pleistocene and mid-Pleistocene climate transitions observed in many proxy records worldwide.

## 1 Introduction

Rivers erode Earth's landscapes and transport bedrock- and biosphere-derived sediment from continents to oceans. Tectonics and climate modulate these processes by building topography on which channels evolve, and by setting discharge via precipitation (e.g., Perron, 2017). Since ~5 Ma, plate tectonic rates remained steady but global climate has varied profoundly (Peizhen et al., 2001; Molnar, 2004). Pliocene–recent global benthic $\delta^{18}O$ records net cooling and glacial–interglacial climate cycles that increased in amplitude and duration at ~2.6 Ma under Pliocene-Pleistocene intensified northern hemisphere glaciation, and at ~1 Ma during the mid-Pleistocene transition from ~40 to ~100 kyr-long cycles (Ahn et al., 2017; Lisiecki and Raymo, 2005). During these climate cycles, intermittent glaciations accelerated erosion in some high-elevation and/or

high-latitude settings (Herman et al., 2013; Willett et al., 2021) while preserving landscapes in others (Bierman et al., 2014; Thomson et al., 2010). In contrast, changes in precipitation coupled to climate cyclicity at ~2.6 Ma and ~1 Ma may have broadly increased variability in runoff and thus amplified river discharge, erosion, and sediment transport in many settings (Peizhen et al., 2001; Molnar, 2004; Bender et al., 2020; Godard et al., 2013).

Multiple archives record continental denudation associated with late Cenozoic climate change. Although susceptible to biases (e.g., Sadler, 1981), many basins worldwide record increased terrigenous sedimentation since ~2–4 Ma (Peizhen et al., 2001; Molnar, 2004) concurrent with increased mountain erosion inferred from thermochronometry (Herman et al., 2013; Willett et al., 2021). Elevated continental denudation rates are also evident in marine isotopic proxies that demonstrate (Misra and Froelich, 2012; Torres et al., 2014) or permit (Li et al., 2021; Willenbring and Von Blanckenburg, 2010) enhanced silicate weathering since ~5–10 Ma. Researchers have long debated whether such enhanced silicate weathering, associated with physical erosion in active mountain ranges where rock uplift rapidly replenishes fresh minerals at Earth's surface, consumed atmospheric $CO_2$ sufficiently to amplify global cooling patterns (Hilton and West, 2020). However, silicate weathering is subordinate to physical erosion only at low erosion rates (West et al., 2005), and denudation of common sedimentary rocks increases net $pCO_2$ (Bufe et al., 2021; Torres et al., 2014). Hence, erosional feedbacks on climate require carbon sequestration mechanisms independent of chemical weathering.

One such sequestration mechanism is the fluvial export of organic carbon from terrestrial landscapes to ocean sediment (Burdige, 2005; Galy et al., 2007, 2015; Hilton et al., 2015). Unlike silicate weathering (West et al., 2005), physical erosion by rivers directly controls terrigenous carbon sequestration in ocean sediment (Galy et al., 2015; Hilton et al., 2015; Hilton and West, 2020) where ~30% of global buried carbon is terrestrially sourced (Burdige, 2005). Himalayan erosion, for example, drives offshore carbon burial far exceeding the system's potential carbon uptake by weathering (Galy et al., 2007), but tectonic rock uplift so dominates Himalayan denudation (Godard et al., 2014) that rate changes from late Cenozoic climate-enhanced runoff may not be detectable (Lenard et al., 2020). Elucidating such changes therefore requires a source-to-sink system more sensitive to climate than tectonics.

Fluvial erosion also controls the riverine supply of terrestrial nutrients (Terhaar et al., 2021; Cotrim da Cunha et al., 2007; Buesseler, 1998) to marine ecosystems, which represent a prominent global $CO_2$ sink (Falkowski et al., 1998). Increased river erosion boosts terrigenous nutrient export to the oceans (Cotrim da Cunha et al., 2007) and thus net primary productivity, which can elevate to >50% the typically small fraction of carbon sequestered via sinking and burial, termed net export production (NEP) (Buesseler, 1998; Falkowski et al., 1998). Nutrients supplied by rivers and coastal erosion support 28–51% of contemporary productivity in the Arctic Ocean (Terhaar et al., 2021), for example. While river erosion modulates the carbon cycle on short geologic timescales by exporting terrestrial organic carbon and nourishing ocean productivity, opportunities to study these mechanisms across late Cenozoic climate changes are rare.

In this paper we use Pliocene–recent records of landscape erosion and marine sedimentation, preserved in terraces along several Yukon River tributaries and in Bering Sea sediment cores, to elucidate links among tectonically quiescent river incision, carbon export, and atmospheric $CO_2$ drawdown across profound global climate changes at 2.6 and 1 Ma.

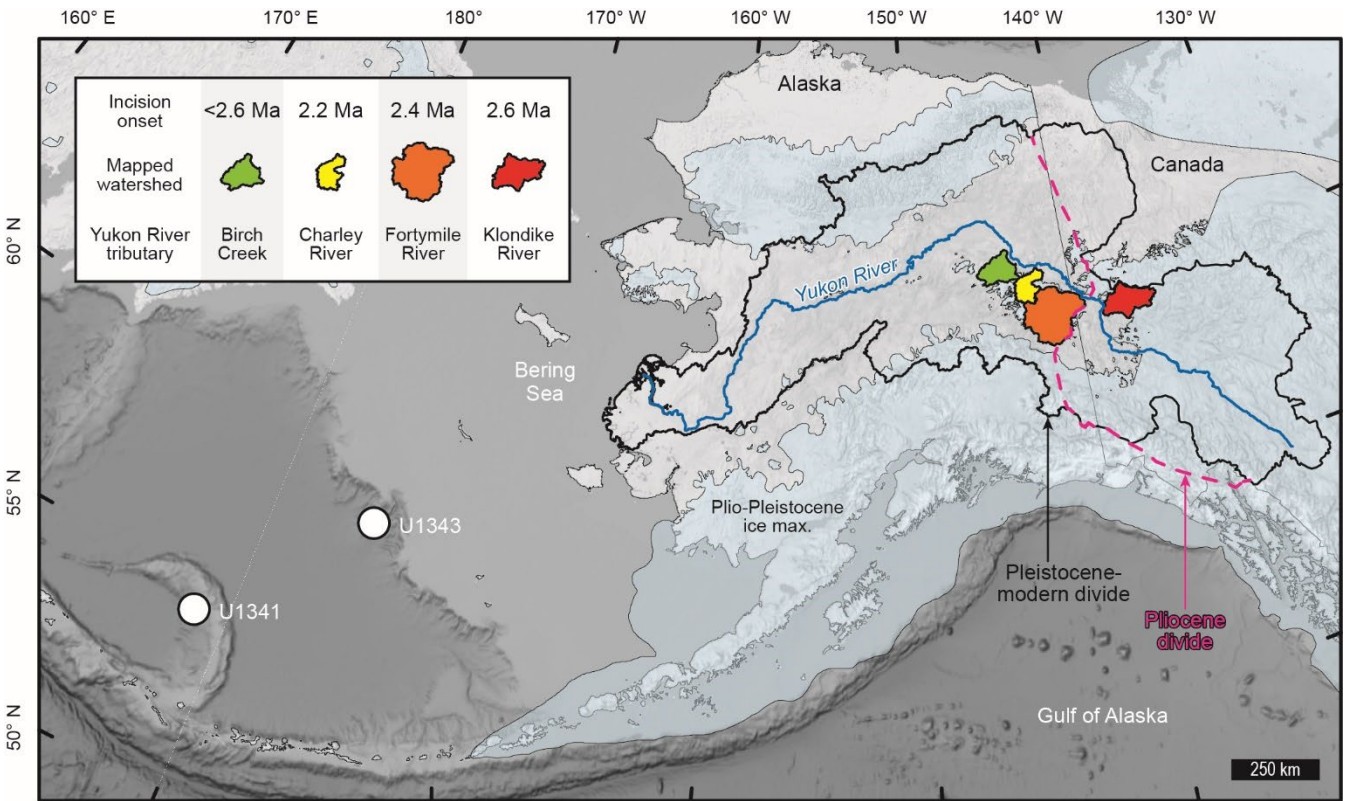

**Figure 1:** The Yukon River-Bering Sea system. Overview depicts the modern (black line) and Pliocene (magenta line; Duk-Rodkin et al., 2001) Yukon River divide, Plio-Pleistocene maximum ice extent (Kaufman et al., 2011; Duk-Rodkin et al., 2001), Integrated Ocean Drilling Program (IODP) core sites (März et al., 2013; Horikawa et al., 2015; Iwasaki et al., 2016; Onodera et al., 2016; Kim et al., 2016). Birch Creek (Ager et al., 1994), Klondike (Hidy et al., 2013; Westgate et al., 2003; Hidy et al., 2018; Lowey, 2006), Fortymile (Bender et al., 2020), and Charley River watersheds colored by incision onset timing. Base map is the General Bathymetric Chart of the Oceans (GEBCO) grayscale rendering (https://www.gebco.net/data_and_products/gridded_bathymetry_data/gebco_2021/).

## 2 The Yukon River-Bering Sea system

The Yukon River-Bering Sea system enables source-to-sink study of linked fluvial erosion, marine sedimentation and carbon deposition, and ocean productivity during late Cenozoic climate changes (Fig. 1). The Yukon River has dominated terrestrial Bering Sea input (Brabets et al., 2000) since ~2.6 Ma, when Cordilleran icesheet expansion dammed the formerly south-flowing headwaters, which subsequently cut northwest across the Pliocene divide, thereby entrenching the modern river course and increasing drainage area ~30% (Duk-Rodkin et al., 2001). Sedimentary records at Integrated Ocean Drilling Program site U1341 archive the 4.3 Ma–recent (März et al., 2013; Iwasaki et al., 2016; Wehrmann et al., 2013; Horikawa et

al., 2015) evolution of the Bering Sea, one of Earth's most productive marine ecosystems. Cores at U1341, collected at 2177 m water depth ~600 km from the Bering Sea shelf, preserve changes in sediment accumulation rate, provenance, and mass proportions of total organic carbon and biogenic silica consistent with a shorter 2.4–1.25 Ma record at site U1343 near the shelf (Kim et al., 2016).

Bering Sea sedimentation changes occurred during global climate transitions at ~2.6 and ~1 Ma, synchronous with pulses of continental river incision archived in two well-preserved strath terrace levels (T1 and T2) up to ~260 m above a prominent Yukon River tributary, the Fortymile River (Bender et al., 2020). Similar terraces flank numerous central Yukon River tributaries east and west of the ancestral Pliocene Yukon River divide (Fig. 1), attesting to widespread river incision likely forced by latest Cenozoic climate change both indirectly [i.e., by icesheet-triggered Yukon River crossing of the Pliocene divide at 2.6 Ma (Duk-Rodkin et al., 2001; Bender et al., 2020)] and directly [i.e., by mid-Pleistocene transition-amplified precipitation and runoff at ~1 Ma (Godard et al., 2013)]. Glaciation restricted to high elevations (Kaufman et al., 2011) preserved terraced tributary landscapes along the central Yukon River, where post-Eocene Tintina Fault quiescence (Bacon et al., 2014) and predominantly Paleozoic and Mesozoic crystalline bedrock (Brabets et al., 2000) define a tectonically quiescent late Cenozoic erosional system more sensitive to climate than rock uplift or erodibility.

Here, we report the previously unknown Pleistocene incision history of the Charley River (Fig. 2). These data, along with previously documented erosion histories in other Yukon River tributaries, demonstrate erosion across at least 60,000 km$^2$ of the central Yukon River basin coupled to carbon burial and paleo-productivity in the Bering Sea during late Cenozoic periods of global climate change.

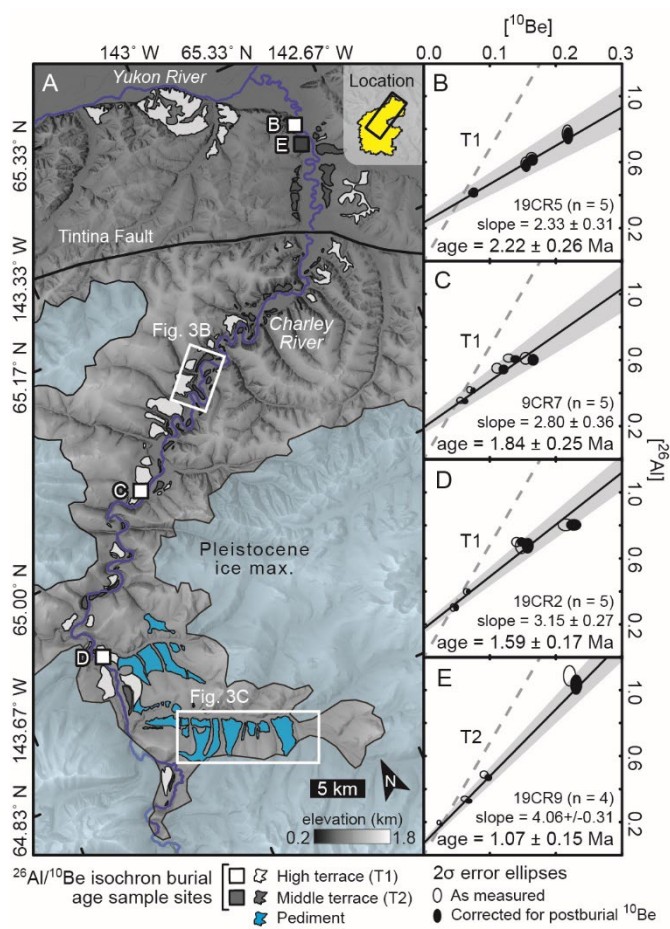


**Figure 2:** Charley River field area and cosmogenic results. (A) Charley River geomorphic map, terrace cosmogenic $^{26}Al/^{10}Be$ isochron burial age sample sites, and Pleistocene ice maximum (Kaufman et al., 2011; Duk-Rodkin et al., 2001) rendered on the IfSAR Alaska digital elevation model (available by searching https://earthexplorer.usgs.gov/). White outlined boxes indicate areas photographed in Figs. 3b–c. (B–E) Cosmogenic $^{26}Al/^{10}Be$ isochron burial age plots, isotope concentrations are in units of $10^6$ atoms g$^{-1}$ quartz. Grey dashed line is the $^{26}Al/^{10}Be$ surface production ratio and ~zero age, grey envelope is the $1\sigma$ regression uncertainty.

## 3 Methods

### 3.1 Mapping and field measurements

Field mapping and digital topography analysis underpin the cosmogenic isotope-constrained Charley River incision history we report herein. At cosmogenic isotope sample sites (Fig. 2) we measured gravel thicknesses and terrace tread heights above the Charley River using a tape measure and TruPulse 200L laser range finder. We mapped high and intermediate terrace levels (T1 and T2, respectively) and the Charley River channel on the 5 m pixel$^{-1}$ Alaska IfSAR digital elevation model (Fig. 2) and extracted elevations from this dataset using ArcMap version 10.8 (https://www.esri.com/en-us/arcgis/products/arcgis-desktop/resources) to develop elevation profiles of Charley River data (Fig. 3a). We also mapped a low-relief surface that is broader, geomorphically higher, slopes more steeply toward valley axes than T1 and T2, occurs

below and adjacent to the maximum Pleistocene ice extent in the Charley River headwaters, and lacks gravel exposures (Figs. 2 and 3). We interpreted this surface as a periglacial pediment (e.g., French, 2016), consistent with the presence of comparable low-relief cold-climate landforms at higher elevations in this environment (e.g., Nyland and Nelson, 2020).

## 3.2 Cosmogenic $^{26}Al/^{10}Be$ isochron burial dating

Along the Charley River (Figs. 2 and 3), a Yukon tributary, we used the cosmogenic $^{26}Al/^{10}Be$ isochron burial method to
(Balco and Rovey, 2008; Zhao et al., 2016) date the latest deposition (and thus earliest incision) of 5–8 m-thick river sediments atop high (T1; three dates) and intermediate (T2; one date) terrace levels mapped up to ~150 and ~30 m above the modern channel, respectively. The isochron method requires sampling quartz-bearing sediment (i.e., cobbles, pebbles, sand) buried by several meters of stratigraphically continuous sediment (indicative of rapid burial deep enough to suppress isotope production and hence initiate decay) at a single depth horizon (indicative of common burial history). The slope of a line fit to
measured $^{26}Al$ and $^{10}Be$ concentrations in quartz from these samples reflects the post-burial isotope decay from the surface production ratio, commonly approximated as 6.8 $^{26}Al/^{10}Be$ atoms though the actual ratio may vary spatially (e.g., with latitude; Halsted et al., 2021), and can therefore be used to calculate the burial duration of the sampled horizon (e.g., Balco and Rovey, 2008; Zhao et al., 2016).

We designed our sample strategy to directly compare results with the previously developed Fortymile River terrace
chronology (Bender et al., 2020). Along the Charley River we sampled three sites on T1 (Figs. A1–A3) to test whether the terrace age decreases upstream, and sampled one T2 site (Fig. A4) to determine if the terrace dates to the 0.7–1.2 Ma mid-Pleistocene climate transition. At each of the four field sites we collected quartz-rich terrace alluvium samples comprising individual cobbles and one several-kilogram sample each of amalgamated pebbles and matrix sand in hand-dug pits from horizons up to 50 cm-thick at depths of 5–7 m below terrace treads. Individual samples ideally yield ~25 grams of pure
quartz for laboratory processing (Corbett et al., 2016); cobble sizes and sand/pebble sample volumes were selected by modal estimation of quartz content to meet or exceed this target mass.

We prepared five samples from each site at the University of Vermont (Corbett et al., 2016); one sample failed to yield sufficient quartz, however, leaving a total of 19 samples. Sample preparation involved crushing and/or sieving each sample to the medium sand size, isolating pure quartz via progressive acid etching and iterative purity testing by laser ablation-
inductively coupled plasma mass spectrometry, and extracting $^{26}Al$ and $^{10}Be$ via column chromatography (full methods available online at https://www.uvm.edu/cosmolab/methods.html). We measured $^{26}Al/^{27}Al$ and $^{10}Be/^9Be$ ratios from each of the 19 samples at the Purdue Rare Isotope Measurement Laboratory (Nishiizumi, 2004; Nishiizumi et al., 2007), corrected each measurement for backgrounds by subtracting blank measurements, and propagated the standard deviation of the blank measurements into sample uncertainties.

The burial isochron is a linear model fit to measured sample nuclide concentrations and analytical uncertainties, with $^{10}$Be and $^{26}$Al on the x- and y-axes, respectively (Balco & Rovey, 2008; Zhao et al., 2016). Samples with a common pre-burial history contain $^{10}$Be and $^{26}$Al concentrations that record post-burial decay from the surface production $^{26}$Al/$^{10}$Be ratio; a line thusly fit to these concentrations can be used to both (a) quantify the duration of post-burial decay (i.e., the burial age) and (b) identify and omit outlier samples with dissimilar pre-burial history. We use the isochron approach of Zhao et al. (2016)

that applies a linearization factor to correct post-burial production among the $^{10}$Be concentrations; the y-intercept reflects post-burial isotope production, but the linearization of $^{10}$Be preserves the slope associated with decay of the inherited pre-burial concentrations. All 19 of the isotope concentration pairs plotted within $2\sigma$ of the regression lines, hence none were omitted from the final linear fits (Fig. 2b–e).

### 3.3 Bering Sea Sedimentation

We combined published age-depth data (Horikawa et al., 2015; Onodera et al., 2016; Iwasaki et al., 2016) to modestly revise an age-depth model that closely matches that of Horikawa et al. (2015) for the Integrated Ocean Drilling Program site U1341 in the Bering Sea (Fig. 4). To produce this model, we visually identified linear segments (n = 5) in the plotted age and depth data, and fit ordinary least squares regression models to each segment. Because age and depth uncertainties are not

consistently reported with the data, we used regression standard errors to quantify uncertainty in the calculated age-depth relationships. We then used these relationships to assign linear sedimentation rates (LSR, cm kyr$^{-1}$) based on slope $m$ of the linear fit segments, and to convert sediment constituent weight percent data depths to ages. We model ages up to 4.3 Ma for sediment TOC (total organic carbon, weight %), weight % $Al_2O_3/SiO_2$ and $Si_{xs}$ [biogenic silica, defined as weight % $SiO_2$ exceeding Upper Continental Crust standard (März et al., 2013)] and detrital $\varepsilon_{Nd}$ (Horikawa et al., 2015) measured in core

U1341. Detrital $\varepsilon_{Nd}$ and $Al_2O_3/SiO_2$, which varies inversely with central Yukon River rock input (Horikawa et al., 2015) and increases with continentality (März et al., 2013), respectively, track sediment provenance.

We estimate the proportions of terrestrial and marine organic carbon in Bering Sea sediment using molar C/N ratios from TOC and N measured in core U1341 (Kim et al., 2016). This approach approximates organic matter provenance crudely due

in part to the wide range of C/N values reported in either environment (Lamb et al., 2006), and because degraded land- and marine-derived particulate organic matter in sediment can yield similar C/N ratios (e.g., Thornton & McManus, 1994). Although higher terrigenous organic sediment fractions likely occur on the Bering Sea shelf near the Yukon River outlet, deep-water molar C/N ratios imply both terrestrial and marine TOC sources since 4.3 Ma. Low C/N molar ratios that average 7.3 in deep-water sites U1341 and U1343 (Kim et al., 2016) imply organic matter predominantly (~85%) derived from

marine NEP based on endmember molar C/N ratios of 5.4 and 19 for marine and terrestrial organic matter, respectively (Perdue and Koprivnjak, 2007). Alternatively, discharge-weighted measurements of particulate organic carbon and nitrogen taken between 2003 and 2012 set an endmember C/N molar ratio of 11.3 for Yukon River suspended sediment (McClelland

et al., 2016), and thus indicate a higher average proportion of terrigenous organic carbon (~86%) assuming the 5.4 marine endmember ratio.


We compute mass accumulation rates (MARs, g cm$^{-2}$ kyr$^{-1}$) for TOC and Si$_{xs}$ as the product of constituent weight percent, sediment mass (g cm$^{-3}$), and LSR. The MAR of Si$_{xs}$ reflects the burial rate of silicate primary producers and thus provides a relative measure of carbon-sequestering NEP (Falkowski et al., 1998); TOC MAR brackets the rate of carbon burial (Wehrmann et al., 2013). Because TOC, Si$_{xs}$, and mass were not measured at the same depth intervals, we used the arithmetic mean of shipboard dry grain density (2.34 g cm$^{-3}$) and assign an uncertainty equivalent to one half the range of shipboard wet sediment masses (0.84 g cm$^{-3}$) in this calculation. We assumed negligible weight percent uncertainty and propagated the LSR and mass uncertainties in quadrature to estimate mass accumulation rate error. MARs represent the burial rate of each constituent and differ from MARs previously estimated for the same Si$_{xs}$ (Iwasaki et al., 2016; März et al., 2013) and TOC (März et al., 2013; Wehrmann et al., 2013) data using shipboard age-depth models that preceded the availability of the full age-depth dataset we employ.

## 4 Results and discussion

### 4.1 Charley River terrace chronology

We report cosmogenic isochron burial ages of Charley River terraces (Figs. 2 and 3) that decrease upstream on high terrace T1 from 2.2 ± 0.3 Ma near the Yukon-Charley River confluence, 1.8 ± 0.3 Ma ~85 km upstream, and 1.6 ± 0.2 Ma ~30 km further upstream. Terrace T1 tread elevations above the modern river also decrease monotonically upstream at sample sites, from 141 m to 107 m to 59 m. Intermediate terrace T2 yields a cosmogenic isochron burial age of 1.1 ± 0.2 Ma near the Charley-Yukon River confluence, ~30 m above the modern river elevation.

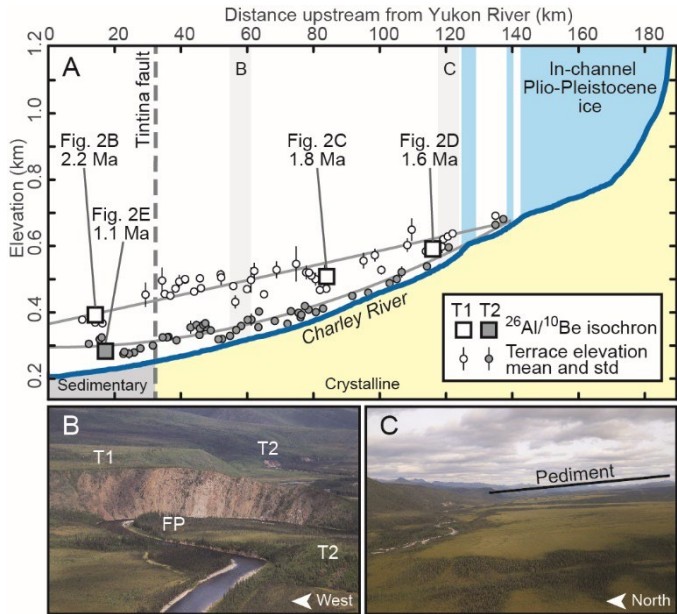

**Figure 3:** Charley River elevation profile and field photos. (A) Alaska IfSAR DEM-derived elevation profiles of the channel, terrace tread (mean and standard deviation), and $^{26}Al/^{10}Be$ isochron burial age sample sites. Light grey lines are quadratic fits to the terrace tread elevations that generalize the paleo-channels. Blue shaded areas above the channel profile indicate river reaches overrun by Pleistocene ice (Kaufman et al., 2011). Letters and light grey bars indicate locations of field photos of (B) river terrace levels T1, T2 and floodplain (FP), and (C) a low-relief surface geomorphically above the river terraces that we interpret as a periglacial pediment (e.g., French, 2016; Nyland and Nelson, 2020).

## 4.2 Climate-driven Yukon River incision

Our Charley River incision chronology closely matches previously established Fortymile (Bender et al., 2020), Klondike River (Westgate et al., 2003; Froese et al., 2000; Hidy et al., 2013; Lowey, 2006), and Birch Creek (Ager et al., 1994) records (Figs. 1 and 4). Charley River terrace tread heights reflect incision depth while burial ages date last fluvial deposition and thus bracket incision onset timing. Terrace height-age data show that Charley River incision propagated ~140 km upstream at ~160 mm kyr$^{-1}$ from 2.2 to 1.6 Ma, stalled during 1.6 to 1.1 Ma as T2 aggraded, and resumed at 1.1 Ma (Fig. 3a). East of the Charley River in the adjacent Fortymile River basin, following sedimentation from 4.8–2.4 Ma, incision propagated upstream through T1 at ~270 mm kyr$^{-1}$ from 2.4 to 1.8 Ma, paused during 1.8–1.1 Ma T2 formation, then resumed at ~1 Ma. Further east and across the Pliocene Yukon River divide, multiple chronometers date Klondike River terraces (Froese et al., 2000; Westgate et al., 2003; Hidy et al., 2013, 2018) east of the Fortymile and Charley Rivers, delineating tens of meters of gravel aggradation on T1 from 4.3 to 2.6 Ma followed by incision that stalled during 1.8–1.1 Ma T2 formation. West of the Charley River, near the Birch Creek-Yukon River confluence, T1 deposits preserve a palynologically constrained mid–late Pliocene sequence of pre- and post-capture gravels (Ager et al., 1994) that permit Pleistocene abandonment, consistent with incision onset timing in the Klondike (2.6 Ma) (Hidy et al., 2013; Westgate et al., 2003; Froese et al., 2000), Fortymile (2.4 Ma) (Bender et al., 2020), and Charley Rivers (2.2 Ma).

Taken together, these terrace chronologies demonstrate climate-modulated pulses of central Yukon River incision that disrupt regional aggradation. Pliocene-Pleistocene icesheet-triggered incision at 2.6 Ma progressed west across the Pliocene divide (Duk-Rodkin et al., 2001) at ~625 mm kyr$^{-1}$ from the Klondike River (2.6 Ma) to downstream of the Charlie River (2.2 Ma). This pulse accelerated widespread tributary erosion across interior Alaska and Yukon until stalling nearly simultaneously on the Klondike (1.8 Ma), Fortymile (1.8 Ma), and Charley Rivers (1.6 Ma) consistent with equilibration of sediment flux and transport capacity set by early Pleistocene climate (Bender et al., 2020; Finnegan et al., 2007). Incision resumed in all tributaries at 1.1 Ma, during the mid-Pleistocene transition from 40-kyr to high-amplitude 100-kyr climate cycles, likely in response to concomitant changes in precipitation, runoff, and fluvial transport capacity (Godard et al., 2013; Finnegan et al., 2007). Klondike (Lowey, 2006), Fortymile (Bender et al., 2020), and central Yukon River (Froese et al., 2005) floodplain deposit ages imply stable channel elevations consistent with negligible incision or aggradation since the latest Pleistocene–Holocene.

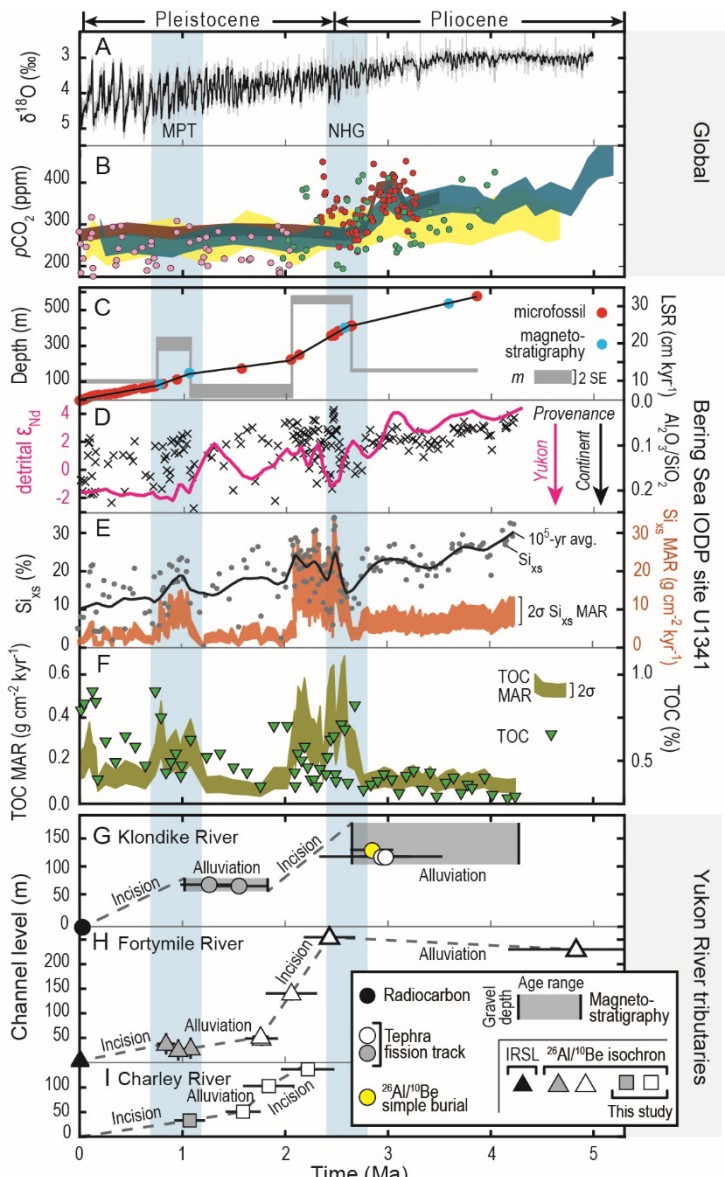

**Figure 4:** Late Cenozoic changes in global climate, Bering Sea sedimentation, and Yukon River tributary incision. (A) Global benthic δ[18]O (black line) and standard deviation (grey bars) (Ahn et al., 2017). (B) Atmospheric $pCO_2$ reconstructed from δ[11]B [red (Martínez-Botí et al., 2015), green (Bartoli et al., 2011), and pink (Hönisch et al., 2009) dots; blue envelope (Seki et al., 2010)] and alkenones [brown (Seki et al., 2010) and yellow (Pagani et al., 2010) envelopes]. (C) Bering Sea Integrated Ocean Drilling Program (IODP) Site U1341 depth–age data [colored dots (Horikawa et al., 2015; Iwasaki et al., 2016)], regression model (black lines). Thick gray line plots regression model slope $m$ and standard error (SE), which signify sedimentation rate and uncertainty. (D) Detrital $Al_2O_3/SiO_2$ weight % ratio [black x's, higher ratios reflect more continental provenance (März et al., 2013)] and 10[5]-yr moving average ε$_{Nd}$ values [magenta line, lower values reflect more Yukon River provenance (Horikawa et al., 2015)]. (E) Silicate biomass inferred from weight % silica exceeding upper continental crust standard [$Si_{xs}$; gray dots (März et al., 2013)] and 10[5]-yr moving average (black line), and $Si_{xs}$ mass accumulation rate (MAR; orange envelope). (F) Total organic carbon [TOC; inverted triangles (März et al., 2013)] and TOC MAR (olive envelope). (G) Klondike River terrace ages over height above floodplain after composite magnetostratigraphy (Froese et al., 2000), tephrochronology (Westgate et al., 2003), cosmogenic burial dating (Hidy et al., 2013, 2018) and floodplain radiocarbon ages (Lowey, 2006). (H) Fortymile

River terrace ages over height above the mid–late Holocene-aged floodplain (Bender et al., 2020). (I) Charley River terrace ages over height above the modern channel. MPT—mid-Pleistocene climate transition; NHG—northern hemisphere glaciation intensification.

## 4.3 Bering Sea response to Yukon River incision

Pulses of central Yukon River incision, from 2.6–2.2 Ma and at 1.1 Ma, enhanced Bering Sea sedimentation, terrigenous carbon burial, and NEP (Fig. 4). Synchronous with the initial incision pulse, from 2.6–2.1 Ma Bering Sea sediment (März et

al., 2013; Iwasaki et al., 2016; Wehrmann et al., 2013; Horikawa et al., 2015) provenance shifted toward continental Yukon River sources and sedimentation rate tripled, attended by two–four-fold MAR increases that raised $Si_{xs}$ ~10% and doubled TOC from 0.3–0.4% to a maximum of 0.8%. Linking this Bering Sea productivity and carbon burial pulse to concurrent Yukon River incision by the sediment provenance indicators detrital $\varepsilon_{Nd}$ and $Al_2O_3/SiO_2$ provides an alternative explanation to the prior interpretations of North Pacific nutrient leakage as a driver of increased $Si_{xs}$ MAR (i.e., productivity) from 2.6–

2.1 Ma at U1341 (März et al., 2013) and from 2.4–1.9 Ma in the shorter record at U1343 (Kim et al., 2016)(Fig. 1).

Incision of the central Yukon River subsequently transitioned to alluviation (1.8–1.6 Ma to 1.1 Ma) concurrent with deceleration and stabilization of Bering Sea sedimentation (2.1–1.1 Ma). Following the second Yukon River incision pulse, induced by the mid-Pleistocene transition (Ahn et al., 2017; Hönisch et al., 2009), the rate of Bering Sea sedimentation

doubled between 1.1 and 0.8 Ma. Sediment continentality and Yukon provenance both sharply increased during this interval, and MARs of $Si_{xs}$ and TOC doubled in association with ~10% and ~two-fold increases in these constituents, respectively. Consistent timing among these records strongly suggests that Yukon River incision and sediment export increased Bering Sea carbon sequestration both by burial of terrestrial organic carbon and by boosting marine NEP during global climate changes at ~2.6 and ~1 Ma. Despite the potential ambiguity of C/N ratios in distinguishing organic matter provenance (e.g.,

Thornton & McManus, 1994), such ratios imply up to 40% terrigenous organic matter from 2.4–2.0 Ma at U1343 near the shelf (Kim et al., 2016), consistent with our C/N ratio-based interpretation of mixed terrestrial and marine organic sediment sources since 4.3 Ma at U1341.

## 4.4 Erosion-climate feedbacks?

Coupled terrestrial erosion and marine carbon sequestration like that we document in the Yukon River-Bering Sea system during global climatic changes at 2.6–2.1 Ma and 1.1–0.8 Ma may help explain concurrent atmospheric $p$CO$_2$ decreases (Fig. 4). Such hydroclimate-dependent fluvial carbon sequestration mechanisms may have occurred in river systems worldwide with similar timing (Molnar, 2004; Peizhen et al., 2001), collectively drawing down atmospheric carbon over timescales comparable to the $10^5$-yr fluvial responses we document. At ~2.4 Ma, amidst the first Yukon-Bering pulse, both the

maximum and range of atmospheric $p$CO$_2$ reconstructed from $\delta^{11}$B (Martínez-Botí et al., 2015; Bartoli et al., 2011; Seki et al., 2010; Hönisch et al., 2009) and alkenones (Seki et al., 2010; Pagani et al., 2010) in ocean sediments at a range of

latitudes decreased >100 ppm. Some but not all $p$CO$_2$ records dip during the second pulse, with a decrease of 25–30 ppm (Seki et al., 2010; Hönisch et al., 2009). Decreased $p$CO$_2$ during these late Cenozoic climate changes may reflect enhanced terrestrial carbon burial and marine NEP driven by accelerated fluvial erosion in systems like the Yukon River. In this case, fluvial response times may ultimately pace source-to-sink export and thus influence subsequent cascading oceanic and climatic changes.

## 5 Conclusions

We report a Charley River terrace chronology that, together with independently established Yukon River tributary histories (Bender et al., 2020; Froese et al., 2000; Hidy et al., 2013; Lowey, 2006; Westgate et al., 2003), demonstrates widespread climate-induced incision at ~2.6 Ma and 1.1 Ma across >60,000 km$^2$ of the central Yukon River basin. Simultaneous with these two incision pulses, pulses of Yukon River-derived continental sediment enhanced both TOC (~85% marine) and Si$_{xs}$ burial in the Bering Sea during 2.6–2.1 Ma and 1.1–0.8 Ma. This coupling strongly suggests that Yukon River incision controlled terrigenous organic carbon and nutrient export to the Bering Sea, and thus carbon sequestration via burial and marine NEP. Global late Cenozoic climate changes may have imposed comparable river erosion responses across Earth's surface (Godard et al., 2013; Molnar, 2004; Peizhen et al., 2001); if true, these hydroclimate-sensitive but mineral weathering-independent mechanisms may help explain repeated $p$CO$_2$ drawdown documented globally (Martínez-Botí et al., 2015; Bartoli et al., 2011; Seki et al., 2010; Hönisch et al., 2009; Pagani et al., 2010).

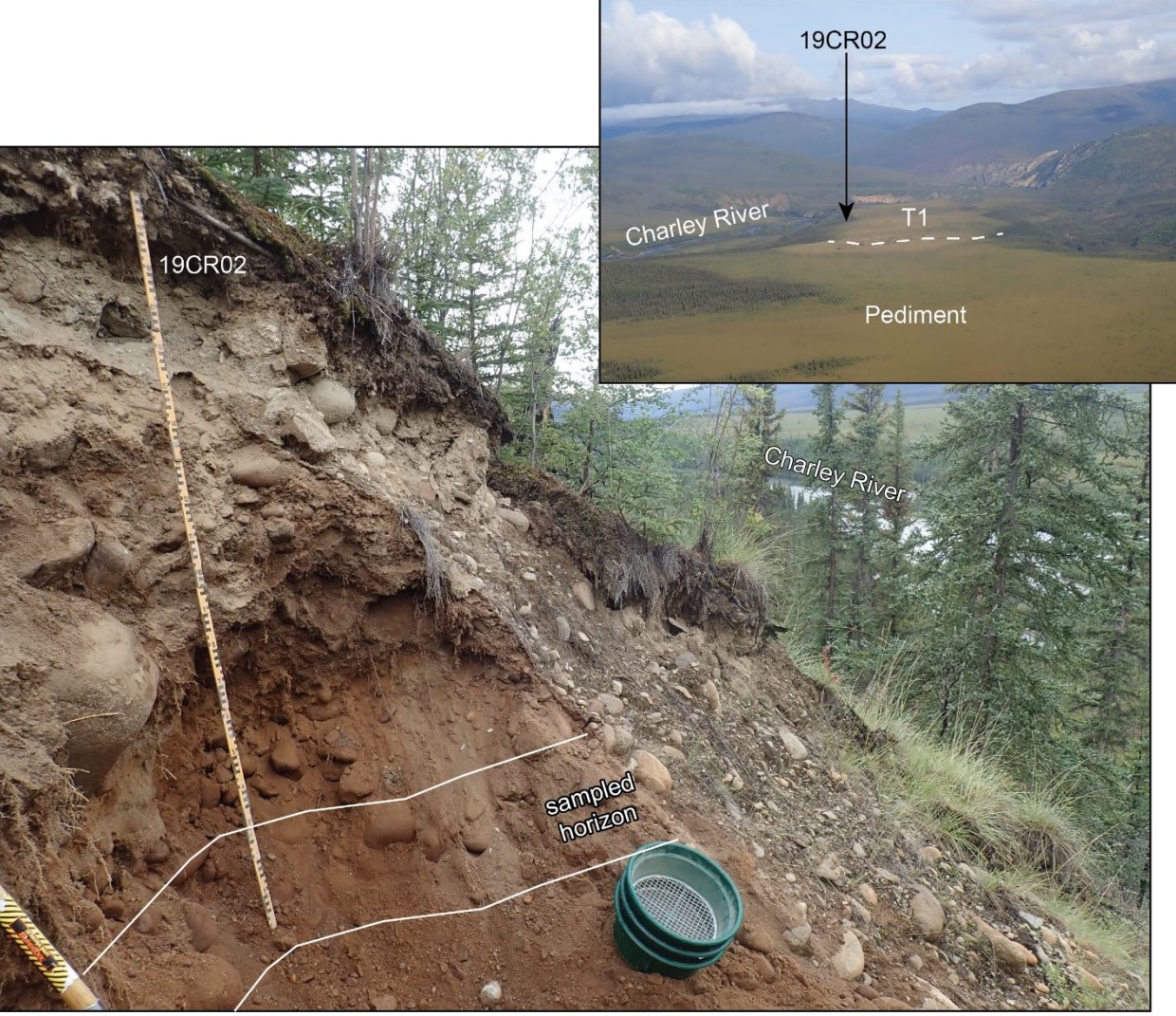

**Figure A1. Cosmogenic isochron burial age sample site 19CR2.** Site located on the Charley River high terrace (T1) in Alaska near the headwaters. Main photo is taken looking upstream (south), inset aerial overview is taken looking downstream (north-northwest).

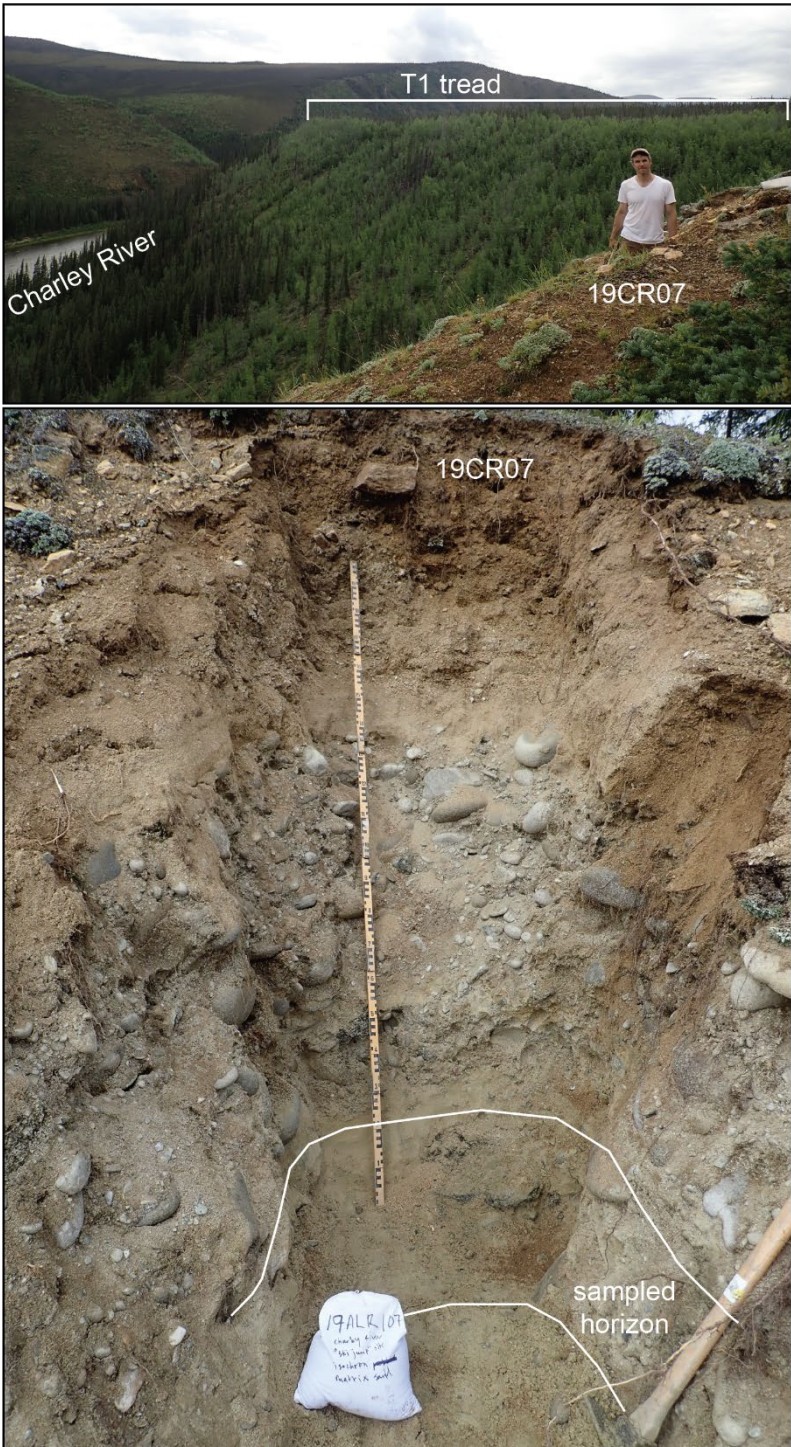

**Figure A2. Cosmogenic isochron burial age sample site 19CR07.** Site located on the Charley River high terrace (T1) in Alaska ~85 km upstream of the Charley-Yukon River confluence. Top photo is taken looking upstream (south-southeast), person is standing in sample pit.

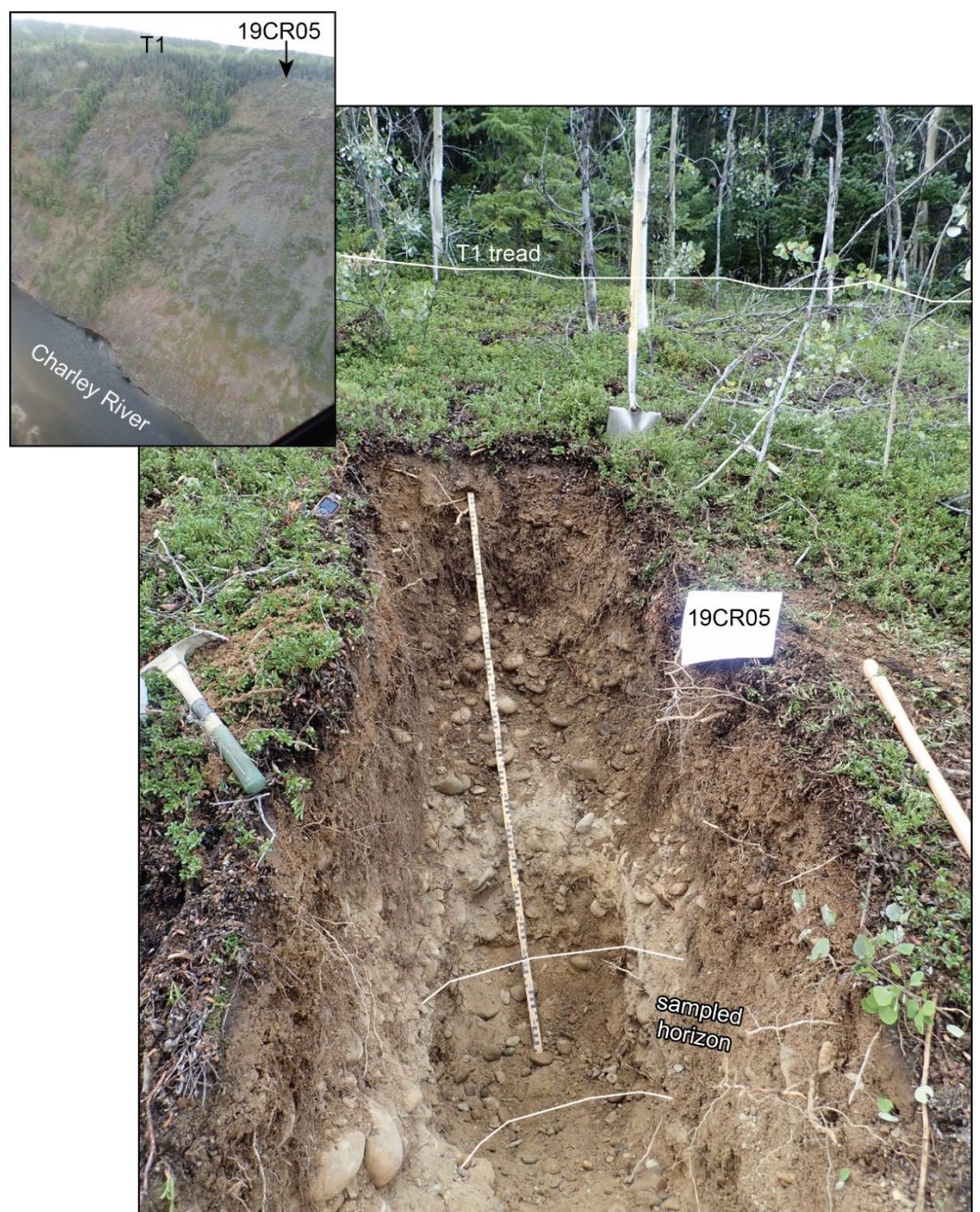

**Figure A3. Cosmogenic isochron burial age sample site 19CR05.** Site located on Charley River high terrace (T1) in Alaska near the Charley-Yukon River confluence. Inset aerial photo is taken looking southwest; Charley River flow is to the right.

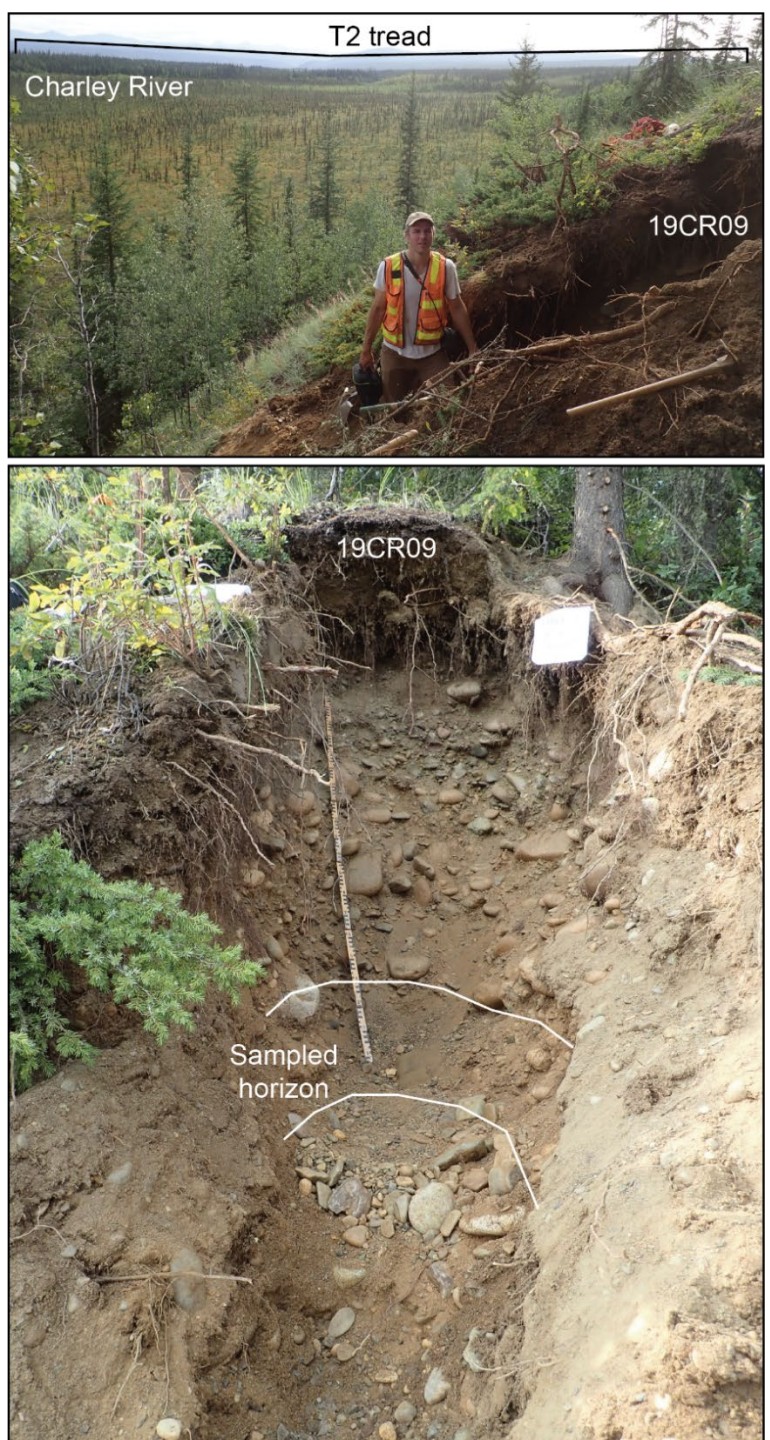

**Figure A4. Cosmogenic isochron burial age sample site 19CR09.** Site located on the Charley River intermediate terrace (T2) in Alaska near the Charley-Yukon River confluence. Top photo is taken looking upstream (south), person is standing in sample pit, the Charley River flows between the broad floodplain in the foreground (bright green) and T2 in the background (dark green).

335

**Data availability:**

Charley River cosmogenic isotope data interpreted in this study are available through the U.S. Geological Survey Alaska Science Center Science Portal with the identifier doi.org/10.5066/P9DRHQIS and are also provided in a supplemental Excel file accompanying this paper.

Fortymile River terrace age data are available through the U.S. Geological Survey Alaska Science Center Science Portal with the identifier doi.org/10.5066/P9XVMTAK and are also provided in a supplemental Excel file accompanying this paper.

The IfSAR-based Alaska digital elevation model is available by searching https://earthexplorer.usgs.gov. Bering Sea sediment data are available by searching https://web.iodp.tamu.edu/OVERVIEW/ and are also provided in a supplemental Excel file accompanying this paper.

Reconstructed $pCO_2$ data and Klondike River data are available in supplemental files linked to the cited publications and are also provided in a supplemental Excel file accompanying this paper.

**Author contributions:**

AB and RL conceptualized the research and conducted field investigations; AB, LBC, PB, and MC conducted laboratory investigations; AB and LBC curated and formally analyzed the cosmogenic isotope data; JVJ, DK and PB acquired funding and resources; AB compiled and analyzed legacy Bering Sea and Yukon River tributary terrace data, wrote the paper, and developed the figures; all authors contributed to editing and revision.

**Competing interests.** The authors declare that they have no conflict of interest.

**Disclaimer.** Any use of trade, firm, or product names is for descriptive purposes only and does not imply endorsement by the U.S. Government.

**Acknowledgements.** We thank the National Park Service and Yukon-Charley Rivers National Preserve for permitting our fieldwork. Thanks to Charlie Bacon and William Craddock for constructive feedback on an early version of this paper, to associate editor Robert Hilton, and to reviewers Jesse Zondervan and Sophie Hage.

**Financial support.** The USGS Mineral Resources Program and U.S. National Science Foundation grants EAR-1735676 to P.R.B. and EAR-0919759 to M.W.C. funded this work.

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
