# Peer review of "Yukon River incision coupled to CO2 drawdown during late Cenozoic climate changes"

_Earth Surface Dynamics, 2022_

## Author Response (AR1)

Jesse Zondervan (Referee comments)
Adrian Bender (Corresponding author replies)

The authors put forward a manuscript demonstrating a link between climate-induced increases in river erosion with sediment and organic carbon export and burial. The incision record is both impressive and appears robust, based mostly on previously published work by the same authors (Nat geo 2020), with some extra data from a third tributary catchment which increases the robustness of the conclusion. I find figure 2 where the authors combine incision data with their compilation of sediment and organic carbon burial convincing, and the careful burial dating and interpretation of terrace age-heigh data is refreshing. It is unlike me to give such a short review because overall, I think this manuscript is well written and has a clear message and a good dataset to support it. The message itself is well thought-out throughout the manuscript and in the introduction in particular. I agree with the authors that the mechanism of organic carbon erosion and burial is an important sink in the geological carbon cycle, perhaps even more so than silicate weathering. I do not have any comments for revision. I am impressed. I look forward to citing this work in my current manuscript.

Jesse Zondervan

The author team and I are honored by this rare review, and thank the reviewer for the thoughtful and supportive feedback. We have made no manuscript revisions in response.

Sophie Hage (Referee comments)
Adrian Bender (Corresponding author replies)

**General Comments**

It was a pleasure to read the study by Bender et al. who provide a new understanding of particulate and carbon export from the Yukon River to the Bering Sea, using 1) newly dated terraces along the Charley River, a tributary of the Yukon River; 2) previous work published by the same authors. The authors identify a link between Yukon River incision/export, Bering Sea sedimentation and climate, likely explaining $CO_2$ drawdown across the Pliocene/Pleistocene and mid-Pleistocene. The study is timely and will be of interest to a wide range of scientific communities. The paper is clearly written and concise, yet it lacks a bit of context in a few places (although I acknowledge that most of the context is provided in cited references). I find that the paper message is well documented by four clear figures (and a few useful pictures in the appendix). I have a few specific comments that may help improve the clarity of the paper.

The author team and I thank the referee for this thoughtful and thorough review, which led to revisions that we think improve the manuscript. We document revisions that aim to incorporate the helpful comments below.

**Specific comments**

INTRODUCTION: Please state the objectives of the paper more clearly at the end of the Introduction. This can be done in one sentence or two. It will help the reader to see the scope and context of the study upfront. In particular, it seems odd that neither the Yukon River, nor the Bering Sea are mentioned in the Introduction. A bit more context is needed as to why these settings are worthy of investigation.

We conclude the revised introduction, **Lines 63–65,** with:

"In this paper we use Pliocene–recent records of landscape erosion and marine sedimentation, preserved in terraces along several Yukon River tributaries and in Bering Sea sediment cores, to elucidate links among tectonically quiescent river incision, carbon export, and atmospheric $CO_2$ drawdown across profound global climate changes at 2.6 and 1 Ma."

METHODS: It would be useful to describe the aims of each of the methodological approaches carried out. How does $^{26}Al/^{10}Be$ isochron burial methods work (in a few words)?

The Methods Section 3.2 now begins with this revised paragraph, **Lines 117–126**:

"Along the Charley River (Figs. 2 and 3), a Yukon tributary, we used the cosmogenic $^{26}Al/^{10}Be$ isochron burial method to (Balco and Rovey, 2008; Zhao et al., 2016) date the latest deposition (and thus earliest incision) of 5–8 m-thick river sediments atop high (T1; three dates) and intermediate (T2; one date) terrace levels mapped up to ~150 and ~30 m above the modern channel, respectively. The isochron method requires sampling quartz-bearing sediment (i.e., cobbles, pebbles, sand) buried by several meters of stratigraphically continuous sediment (indicative of rapid burial deep enough to suppress isotope production and hence initiate decay) at a single depth horizon (indicative of common burial history). The slope of a line fit to measured $^{26}Al$ and $^{10}Be$ concentrations in quartz from these samples reflects the post-burial isotope decay from the surface production ratio, commonly approximated as 6.8 $^{26}Al/^{10}Be$ atoms though the actual ratio may vary spatially (e.g., with latitude; Halsted et al., 2021), and can therefore be used to calculate the burial duration of the sampled horizon (e.g., Balco and Rovey, 2008; Zhao et al., 2016)."

**Line 90**: I feel that the final findings of the paper are announced too early here

**Now Lines 94–97** We soften the early statement of findings by revising to:

"Here, we report the previously unknown Pleistocene incision history of the Charley River (Fig. 2). These data, along with previously documented erosion histories in other Yukon River tributaries, demonstrate erosion across at least 60,000 km$^2$ of the central Yukon River basin coupled to carbon burial and paleo-productivity in the Bering Sea during late Cenozoic periods of global climate change."

**Line 153:** How did you quantify that 85 % of the organic matter is of marine origin?

Was a mixing model used? Can it be shown/expanded in the text?

Now Lines 155–159 To describe the endmember ratio scheme we used, we revise to:

"Although higher terrigenous organic fractions likely occur on the Bering shelf nearer the Yukon River outlet, deepwater TOC sources are both terrestrial and marine; low C/N ratios that average 7.3 in deep-water sites U1341 and U1343 (Kim et al., 2016) imply organic matter predominantly (~85%) derived from marine NEP based on endmember molar C/N ratios of 5.4 and 19 for marine and terrestrial organic matter, respectively (Perdue and Koprivnjak, 2007)."

**Technical corrections**

**Lines 101-102**: Can you reword this sentence?

Now Lines 105–106 To improve clarity, we revise to:

"Field mapping and digital topography analysis underpin the cosmogenic isotope-constrained Charley River incision history we report herein."

**Lines 120-123:** "We designed our sample strategy to directly compare results with the previously developed Fortymile River terrace chronology (Bender et al., 2020), sampling three sites along T1 (Figs. A1–A3) to test whether the terrace age decreases upstream and one T2 site (Fig. A4) to test whether the terrace age overlaps the 0.7–1.2 Ma mid-Pleistocene climate transition as observed along the Fortymile River."

--> This sentence is hard to follow (e.g. "test whether" is used twice). Can you simplify?

Now Lines 122–126 To simplify, we revise to:

"We designed our sample strategy to directly compare results with the previously developed Fortymile River terrace chronology (Bender et al., 2020). Along the Charley River we sampled three sites on T1 (Figs. A1–A3) to test whether the terrace age decreases upstream, and sampled one T2 site (Fig. A4) to determine if the terrace dates to the 0.7–1.2 Ma mid-Pleistocene climate transition."

**Line 130:** remove "d" from "measured"

To maintain consistency with the rest of the methods section, which is written in past tense, we prefer to not change "measured" to "measure" as suggested.

**Line 134**: add "use" between "We" and "the"

Done, thanks for catching.

**Line 147:** What does "ages up to 4.3 Ma of %TOC" mean?

Now Lines 150–154 Refers to ages modeled for the sediment constituent quantities we interpret in the manuscript. Do decrease ambiguity we revised to:

"We model ages up to 4.3 Ma for sediment TOC (total organic carbon, weight %), weight % $Al_2O_3/SiO_2$ and $Si_{xs}$ [biogenic silica, defined as weight % $SiO_2$ exceeding Upper Continental Crust standard (März et al., 2013)] and detrital $\varepsilon Nd$ (Horikawa et al., 2015) measured in core U1341."

**Lines 182 to 185:** "Charley River terrace tread heights reflect incision depth, and burial ages date last fluvial deposition and thus, approximately, incision onset; these data imply that incision advanced ~140 km upstream at ~160 mm kyr-1 from 2.2 to 1.6 Ma, stalled during 1.6 to 1.1 Ma as T2 aggraded, and resumed at 1.1 Ma"

--> Can you simplify this sentence? Split in two parts?

Now Lines 187–190 To simplify, we revise to:

"Charley River terrace tread heights reflect incision depth while burial ages date last fluvial deposition and thus bracket incision onset timing. Terrace height-age data show that Charley River incision propagated ~140 km upstream at ~160 mm kyr$^{-1}$ from 2.2 to 1.6 Ma, stalled during 1.6 to 1.1 Ma as T2 aggraded, and resumed at 1.1 Ma (Fig. 3a)."

---

## Author Response (AR2)

Bob Hilton (Comments to the author)
Adrian Bender (author responses)

Dear Authors,

Two reviewers have now reviewed your work. I completed my own review independently, and was in agreement with the reviewers. This paper reports new incision rate data from the Charley River (using 10Be-26Al burial isochrons) and interprets it alongside some reanalysis of a Bering Sea sediment core to propose links between timings of Yukon River incision, sediment export, marine and terrestrial organic carbon burial, and the global changes in atmospheric CO2 and climate over this period.

The initially revised version you supplied tackles comments by reviewer #2 (reviewer #1 was overwhelmingly positive and did propose any changes). I have some further minor comments that relate to some of those comments from R2, calling for a few clarifications in places, and some more discussion of the source of organic matter.

I also think that given the nature of the study (one river basin vs global atmospheric composition and climate) and the details of the timings of incision and TOC accumulation (which Figure 4 shows leads in MAR compared to incision) mean that some of the language in the manuscript is a little to "strong". I make some suggestions that retain the interesting message, but more fairly reflect the findings and scale of the study.

The authors and I thank you for your thorough independent review. We document revisions below that attempt to accommodate your suggestions, which clarify and honor the scope of our work.

1: "coupled to CO2 drawdown" may be a bit of a push for the title. Also, its more that you are linking Yukon river incision to marine organic carbon burial, which has implications for the global carbon cycle. Would this not be a fairer link to make in the title? Late Cenozoic could also perhaps be better put (its about the last ~10% of the Cenozoic...), perhaps as "last 3 million years" or similar.

Revised title to: Yukon River incision drove organic carbon burial in the Bering Sea during global climate changes at 2.6 and 1 Ma

22: wording is a bit strong here. I suggest changing "explain" to "contribute to"

Done.

75: can you give the modern day water depth (and perhaps distance from continental shelf) for U1341?
Revised **lines 83–85** to: "Cores at U1341, collected at 2177 m water depth ~600 km from the Bering Sea shelf, preserve changes in sediment accumulation rate, provenance, and mass proportions of total organic carbon and biogenic silica consistent with a shorter 2.4–1.25 Ma record at site U1343 near the shelf (Kim et al., 2016)."

80: what does "pristine" mean? Perhaps rephrase
Now **line 87**, changed "pristine" to "well-preserved."

82: clarify what is the "Pliocene divide" and refer to Figure 1 if that is related.
Revised **lines 88–89** to: "Similar terraces flank numerous central Yukon River tributaries east and west of the ancestral Pliocene Yukon River divide (Fig. 1), …"

123: this is quite a range of grain sizes – can you clarify exactly what was collected and roughly how much mass/volume?
Revised **lines 132–136** to: "At each of the four field sites we collected quartz-rich terrace alluvium samples comprising individual cobbles and one several-kilogram sample each of amalgamated pebbles and matrix sand in hand-dug pits from horizons up to 50 cm-thick at depths of 5–7 m below terrace treads. Individual samples ideally yield ~25 grams of pure quartz for laboratory processing (Corbett et al., 2016); cobble sizes and sand/pebble sample volumes were selected by modal estimation of quartz content to meet or exceed this target mass."

126: In addition to the changes requested by R2, can you please provide more detail on the preparation – I presume a certain grain size was targeted for consistency? Or was the whole sample crushed and processed?
Revised **lines 137–140** to: "We prepared five samples from each site at the University of Vermont (Corbett et al., 2016); one sample failed to yield sufficient quartz, however, leaving a total of 19 samples. Sample preparation involved crushing and/or sieving each sample to the medium sand size, isolating pure quartz via progressive acid etching and iterative purity testing by laser ablation inductively coupled plasma mass spectrometry, and extracting $^{26}$Al and $^{10}$Be via column chromatography (full methods available online at https://www.uvm.edu/cosmolab/methods.html)."

152: In addition to the clarifications called for by R2, it would be useful to note that the C/N ratio is a somewhat crude proxy, in that degraded soil organic matter (and rock organic matter) can have lower C/N ratios that are similar to that of marine phytoplankton. Some discussion of this caveat would be useful. For instance, there are some C/N data published for the Yukon, with McClelland et al., 2016, Global Biogeochemical Cycles, reporting a molar C/N ratio weighted by discharge as 11.7 mol/mol (C/N %/% = 10.0). So the terrestrial input value may be quite a bit lower than used by Kim et al.,

Added a paragraph and a few references to describe C/N approach and limitations, **lines 167–178**: "We estimate the proportions of terrestrial and marine organic carbon in Bering Sea sediment using molar C/N ratios from TOC and N measured in core U1341 (Kim et al., 2016). This approach approximates organic matter provenance crudely due in part to the wide range of C/N values reported in either environment (Lamb et al., 2006), and because degraded land- and marine-derived particulate organic matter in sediment can yield similar C/N ratios (e.g., Thornton & McManus, 1994). Although higher terrigenous organic sediment fractions likely occur on the Bering Sea shelf near the Yukon River outlet, deep-water molar C/N ratios imply both terrestrial and marine TOC sources since 4.3 Ma. Low C/N molar ratios that average 7.3 in deep-water sites U1341 and U1343 (Kim et al., 2016) imply organic matter predominantly (~85%) derived from marine NEP based on endmember molar C/N ratios of 5.4 and 19 for marine and terrestrial organic matter, respectively (Perdue and Koprivnjak, 2007). Alternatively, discharge-weighted measurements of particulate organic carbon and nitrogen taken between 2003 and 2012 set an endmember C/N molar ratio of 11.3 for Yukon River suspended sediment (McClelland et al., 2016), and thus indicate a higher average proportion of terrigenous organic carbon (~86%) assuming the 5.4 marine endmember ratio."

Figure 3B – I wonder if additional annotation might make it a little clearer which horizontal surfaces T1 and T2 refer to?

An earlier draft of this figure featured lines delineating the terrace tread boundaries for the T1 and T2 surfaces. Unfortunately, this additional annotation substantially cluttered the photo and obscured the actual landforms. We agree that more annotation could be useful, but adding it would likely require a much larger format figure than the medium-sized one we have intentionally prepared. Hence, we favor the minimally annotated photo depiction of the terraces in the figure in its current form.

221: this section (and/or previous one about the location of the cores used in this

study) could better explain the caveats associated with interpreting this U1341 record in terms of Yukon inputs alone. And perhaps expand on the links mentioned with U1343 around line 78.

Revised section into two paragraphs. First paragraph describes first incision and sedimentation pulse and concludes with new **lines 251–254**: "Linking this Bering Sea productivity and carbon burial pulse to concurrent Yukon River incision by the sediment provenance indicators detrital $\varepsilon_{Nd}$ and $Al_2O_3/SiO_2$ provides an alternative explanation to the prior interpretations of North Pacific nutrient leakage as a driver of increased $Si_{xs}$ MAR (i.e., productivity) from 2.6–2.1 Ma at U1341 (März et al., 2013) and from 2.4–1.9 Ma in the shorter record at U1343 (Kim et al., 2016)(Fig. 1)."

 221: Also in this section, the source of carbon could be better explained and linked to the results discussion. And how this is different or not from the inferred source of carbon in the U1343 core which is closer to the continental shelf?

Revised section into two paragraphs. First paragraph (above) should partially address this suggestion. Additionally, we conclude the second paragraph with a word about organic matter provenance in Bering Sea cores on new **lines 251–254**: "Despite the potential ambiguity of C/N ratios in distinguishing organic matter provenance (e.g., Thornton & McManus, 1994), such ratios imply up to 40% terrigenous organic matter from 2.4–2.0 Ma at U1343 near the shelf (Kim et al., 2016), consistent with our C/N ratio-based interpretation of mixed terrestrial and marine organic sediment sources since 4.3 Ma at U1341."

231: this final sentence should be split. The last bit is conjecture – the timing is consistent – although one could argue the incision is happening after the burial peak (?), and so the wording here should be more cautious.

Revised **lines 257–259** to: "Consistent timing among these records strongly suggests that Yukon River incision and sediment export increased Bering Sea carbon sequestration both by burial of terrestrial organic carbon and by boosting marine NEP during global climate changes at ~2.6 and ~1 Ma."

252: remove "tight"
Done.

Bob Hilton AE
Oxford, UK